# Study protocol for a randomised controlled trial assessing the impact of pulmonary rehabilitation on maximal exercise capacity for adults living with post-TB lung disease: Global RECHARGE Uganda

Winceslaus Katagira [1], Mark W. Orme [2,3], Amy V. Jones [2,3], Richard Kasiita,[4] Rupert Jones,[5] Andy Barton,[5] Ruhme B. Miah,[2,3] Adrian Manise,[6] Jesse A. Matheson,[7] Robert C. Free,[3] Michael C. Steiner,[2,3] Bruce J. Kirenga,[1] Sally J. Singh[2,3]

► Prepublication history and additional online supplemental material for this paper are available online. To view these files, please visit the journal online (http://dx.doi.org/10.1136/bmjopen-2020-047641).

**Correspondence to**
Dr Winceslaus Katagira; wincegira@gmail.com

## ABSTRACT

**Introduction** The burden of post-tuberculosis (TB) lung disease (PTBLD) is steadily increasing in sub-Saharan Africa, causing disability among TB survivors. Without effective medicines, the mainstay of PTBLD treatment evolves around disease prevention and supportive treatment. Pulmonary rehabilitation (PR), a low-cost, non-pharmacological intervention has shown effectiveness in a group of PTBLD individuals but has not been tested in a clinical trial. This study aims to assess the impact of a 6-week PR programme on maximal exercise capacity and other outcomes among adults in Uganda living with PTBLD.

**Methods and analysis** This is a randomised waiting-list controlled trial with blinded outcome measures, comparing PR versus usual care for patients with PTBLD. A total of 114 participants will be randomised (1:1) to receive either usual care (on the waiting list) or PR, with follow-up assessments at 6 weeks and 12 weeks postintervention. The primary outcome is change in walking distance measured by the Incremental Shuttle Walk Test from baseline to the end of 6 weeks of PR. All secondary outcomes will be compared between the PR and usual care arms from baseline to 6-week and 12-week follow-ups. Secondary outcomes include self-reported respiratory symptoms, physical activity, psychological well-being, health-related quality of life and cost–benefit analysis. All randomised participants will be included in the intention-to-treat analysis population. The primary efficacy analysis will be based on both per-protocol and modified intention-to-treat populations.

**Ethics and dissemination** The trial has received ethical clearance from the Mulago Hospital Research and Ethics Committee (MHREC 1478), Kampala, Uganda as well as the Uganda National Council for Science and Technology (SS 5105). Ethical approval has been obtained from the University of Leicester, UK research ethics committee (Ref No. 22349). Study findings will be published in appropriate peer-reviewed journals and disseminated at appropriate

## Strengths and limitations of this study

► The study aims to determine the effectiveness of pulmonary rehabilitation (PR) for individuals with post-tuberculosis lung disease (PTBLD) in a clinical trial setting. To our knowledge, this is the first pragmatic, fully powered effectiveness trial for PR in PTBLD in Africa. This is a progression of previous work that established feasibility and acceptability of PR design for people living with PTBLD in Uganda.

► Due to funding limitations, we are unable to carry out a multi-site study. This may limit generalisability of the study findings.

local, regional and international scientific meetings and conferences.

**Trial registration number** ISRCTN18256843.
**Protocol version** Version 1.0 July 2019.

## INTRODUCTION

### Background and rationale

In 2018, 24% of the global Tuberculosis (TB) incident cases occurred in the African region.[1] Furthermore, 24 of the 30 high TB/HIV burden countries, including Uganda, are in the African region; accounting for 71% of the global burden of HIV associated TB.[1] Despite great strides made over the recent years to achieve the 90% treatment success rate, as part of the 'End TB strategy' target,[1] a significant number of TB survivors continue to have poor health-related quality of life (HRQoL).[2] This may be attributed to the pulmonary function impairment following TB treatment, which has been reported in

approximately 50% of pulmonary TB survivors.[3] The reduction in ventilation and perfusion attributed to the permanent lung parenchymal damage[4] clinically manifests as long-term respiratory symptoms and eventually chronic respiratory disease (CRD), including chronic obstructive pulmonary disease (COPD), bronchiectasis and aspergillosis.[5 6]

Adults with post-TB respiratory symptoms develop skeletal muscle dysfunction, related to physical inactivity and systemic inflammation, which is often compounded by impaired nutrition and poverty.[7] Such patients enter a vicious cycle with falling body weight, progressive morbidity and increased mortality.[7] Individuals affected by CRDs tend to avoid exercise and become increasingly deconditioned and demotivated, leading to a cycle of decline. There are no effective medicines for post-TB lung disease (PTBLD) and the mainstay of treatment evolves around disease prevention and supportive treatment. The disease, previously neglected by health services and researchers, is now the focus of increasing interest.[8 9]

In low-income and middle-income countries (LMIC) where healthcare focuses on treatment and prevention of infectious diseases, as opposed to managing chronic diseases, the care for adults living with CRD presents a major challenge. Consequently, patients that require long-term and systemic approaches often receive suboptimal medical care, inevitably leading to preventable deaths in resource poor settings.

Pulmonary Rehabilitation (PR) is a low cost, high impact intervention that reverses the disability associated with CRDs, and is supported by the highest level of research evidence in high-income countries.[10 11] A PR programme brings together health professionals from many disciplines offering supervised exercise training and disease education, supporting people to manage their own disease. However, in LMIC where the burden of CRDs is increasing fastest, PR is scarce and healthcare services are poorly adapted to deal with such diseases. Although PR is a grade 'A' evidence treatment for adults with COPD[12] and has been used in other chronic lung diseases,[13] its efficacy in PTBLD is not known. In a development study to examine the impact of PR for people with PTBLD in Uganda, it was feasible to run a PR programme and participants reported clinically important improvements in quality of life, exercise capacity and respiratory outcomes.[14] To date, there has been little attention to the role of PR in PTBLD globally, particularly in Africa where a significant number of PTB survivors reside.

### Study objectives
The primary objective of this trial is to assess the impact of a 6-week PR programme on maximal exercise capacity using the incremental shuttle walking test (ISWT) among adults living with PTBLD postintervention.

The secondary objectives include assessing the impact of PR on quality of life and other outcomes for patients with PTBLD, and to conduct a cost–benefit analysis of PR.

### METHODS
### Study design
This is a prospective, randomised waiting-list controlled trial with blinded outcome measures, comparing PR versus usual care for patients with PTBLD. During this effectiveness trial, a total of 114 participants will be

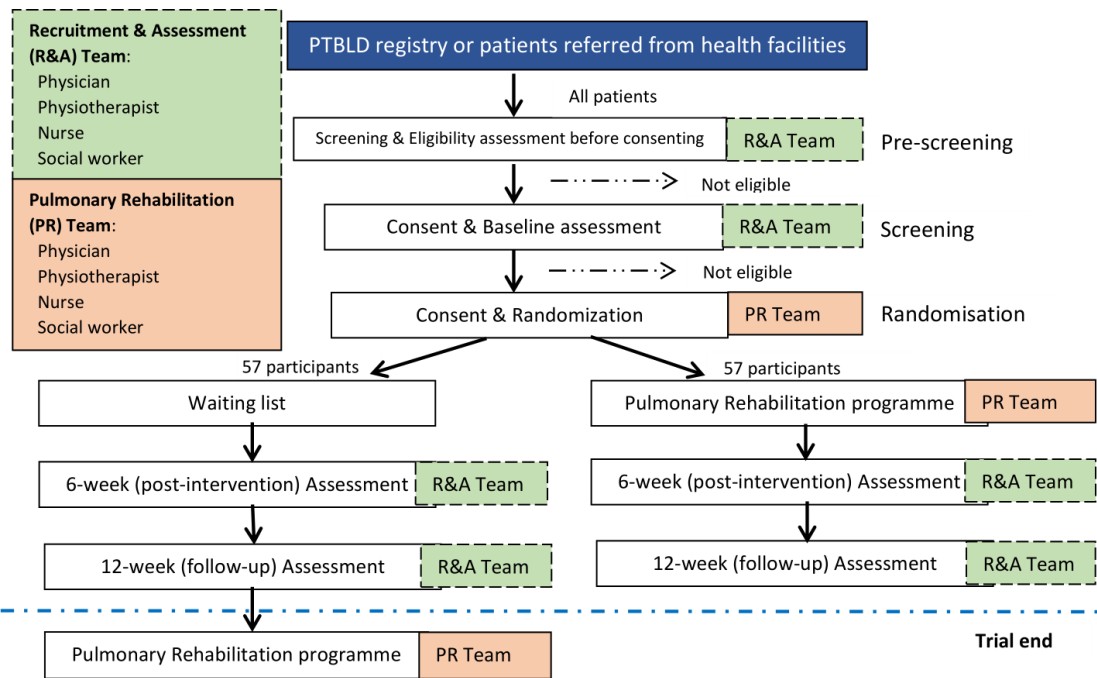

**Figure 1** Figure showing the study flow in the post-TB pulmonary rehabilitation (PR) trial. PTBLD, post-TB lung disease; R&A, recruitment and assessment; TB, tuberculosis.

randomised (1:1) to receive either usual care (waiting-list) or PR (figure 1).

## Study setting

The study is conducted at the PR centre located at the Makerere University Lung Institute (MLI) Clinic, Kampala, Uganda. The MLI clinic is an academic outpatient clinic within the Mulago National Referral hospital, a teaching and clinical research hospital for Makerere University.

## Study population

### Recruitment

Adults with PTBLD will be referred from health facilities and clinics (TB treatment centres and HIV/TB caring centres) around Kampala to the PR centre. Existing registers have around 300 adults living with PTBLD and additional patients will be screened directly from the outpatient departments.

In this study, a patient is considered to have PTBLD if they successfully completed treatment for microbiologically confirmed pulmonary TB but continue to experience chronic respiratory symptoms with radiological evidence of lung parenchymal damage.

### Participant invitation

The process of identifying and inviting eligible patients was refined in the development study. Eligible individuals identified as having an established PTBLD diagnosis will be received at the PR centre at the MLI. Literate participants will be asked to read the patient information sheet (PIS) about the study, written in English or translated in the local language. Illiterate participants will have the contents read to them in full by a study staff, in the presence of a witness who will be present during the whole process. Participants will have the opportunity to discuss the PIS with the study medical personnel. Once the study staff are satisfied that the participant has understood the PIS, and is interested in taking part in the study, they will be taken through the informed consent process. Participants will give consent before undergoing screening tests and procedures, and if still eligible after the screening process, will be taken through another informed consent process for randomisation.

## Eligibility criteria

### Inclusion criteria

A patient with PTBLD is eligible for the trial if they meet all of the following criteria: aged ≥18 years, willing and able to provide written informed consent (signed or witnessed consent if the patient is illiterate), a documented history of smear positive pulmonary TB with treatment completed ≥6 months prior to study enrolment, a negative Xpert MTB/RIF assay for *Mycobacterium tuberculosis* at the time of study enrolment, and report a Medical Research Council (MRC) dyspnoea grade ≥2.

### Exclusion criteria

A PTBLD patient is ineligible for the study if they have co-morbidities that preclude exercise (eg, known unstable cardiovascular disease, locomotor difficulties) or if they are unwilling to participate for any reason or had any condition (social or medical) which in the opinion of the investigator would make study participation unsafe.

## Randomisation

Once eligible participants have consented to take part in the study, they will be randomised using a web-based randomisation system (https://www.sealedenvelope.com/). Participants will be randomised (1:1) to receive either usual care or PR. Access to the web-based system will be controlled through an authorised username and password. Randomisations will be conducted by a member of the study team independent from the data collection team and will be revealed to the data collection and intervention delivery teams after baseline measurements have been obtained.

## Participant timeline

After randomisation, the PR team will explain to participants when the PR sessions will take place. For each individual participant, the hospital based PR programme will last 6 weeks, followed by a follow-up period of 6 weeks of home exercises. Participants in the control arm (waiting-list) of the trial will be informed of the date for their first exercise session in approximately 12–15 weeks. Based on our development study,[15] we expect to find prolonged and possibly improved effects of PR at follow-up. Our experience indicates that a follow-up period of more than 3 months after the start of the PR programme would be unrealistic in this environment without unacceptable attrition. Study participants will receive compensation for their time and transport.

## PR team

The PR team has received adequate training on the delivery of PR and participated in the development study which informed this trial.[14] Furthermore, the individuals are registered health professionals (physiotherapist, physicians and nurses) and have undertaken training regarding the study tests, procedures and measurements per protocol as well as Good Clinical Practice.

## Assessment and follow-up

Participants in both arms of the trial will be asked to attend the baseline, 6-week and 12-week postintervention assessment visits at the PR centre at MLI. Data will be collected by the study staff (medical doctor, nurse and physiotherapist). Table 1 shows all baseline and follow-up assessment data that will be collected during the trial, in accordance with a minimum recommended dataset for PR trials in LMIC.[16]

## Study procedures

During the screening visit, prospective participants will undergo clinical examination, MRC dyspnoea grading, sputum examination using Xpert MTB/RIF assay and a frontal chest radiograph. In addition, demographic, socioeconomic, medical and clinical history (including

**Table 1** The table shows the assessment and follow-up schedule

| Observation/Investigation | Screening/baseline assessments | Randomisation | 12 weeks of study participation | |
|---|---|---|---|---|
| | | | Hospital based pulmonary rehab | Follow-up phase of homebased exercises |
| | | | End of 6 weeks of PR | End of 6 weeks of home exercises |
| Written informed consent | x | x | | |
| Demographics | x | x | | |
| Medical history | x | x | | |
| Clinical exam | x | x | | |
| Chest X-ray | x | | | |
| Spirometry | | x | | |
| MRC dyspnoea grade | x | | x | x |
| Assess symptoms | x | | x | x |
| Incremental Shuttle Walk Test | x | | x | x |
| Endurance Shuttle Walk Test | x | | x | x |
| Borg breathlessness scale | x | | x | x |
| Mid Upper Arm Circumference | x | | x | x |
| Sit-to-stand time | x | | x | x |
| COPD Assessment Test | | x | x | x |
| Clinical COPD Questionnaire | | x | x | x |
| Patient Health Questionnaire | | x | x | x |
| HADS | | x | x | x |
| WPAI | | x | x | x |
| Physical Activity (Actigraph monitor) | | x | x | x |
| Cost/benefit analysis | | x | x | x |
| EQ-5D-5L Questionnaire | | x | x | x |

COPD, chronic obstructive pulmonary disease; EQ-5D-5L, European Quality of Life 5-Dimensions; HADS, Hospital Anxiety and Depression Scale; MRC, Medical Research Council; WPAI, Work Productivity and Activity Impairment.

respiratory symptoms and exposure history to cigarettes and biomass) will be collected using a standardised questionnaire. At the randomisation visit, spirometry will be performed using American Thoracic Society and European Respiratory Society guidelines.[17]

## Sample size
The study will be powered to detect a 35 m difference in the ISWT measured at baseline and after completion of PR.[18] Assuming that ISWT follows an approximately normal distribution, a power calculation based on a paired t-test was performed. Based on a trial sample size of 40 participants in each of the treatment and control groups, a two-sided 5% significance level and a statistical power of 80%, the clinically important change in ISWT of 35 m will also be statistically significant. Our recent feasibility study[15] was used to obtain an estimate of the pooled SD for the power calculation. Conservatively assuming up to 30% losts to follow-up at 6 weeks, a total of 114 participants are required to be recruited and randomised (1:1) to each arm (PR: 57 participants or waiting list: 57 participants). Using the 70% ineligibility rate during screening from the feasibility study, we will need to screen approximately 543 PTBLD patients.

## Blinding (masking)
Due to the nature of PR, it will not be possible to blind participants to their group allocation but participants will be asked not to reveal their group during the follow-up assessments. The participant and treating clinician will be aware of treatment allocation, however, the outcome measures will be performed by staff blinded to treatment allocation and the ISWT (primary outcome) will be prioritised to reduce the risk of un-blinding. Any episodes of unblinding will be documented and reported.

## Treatment arms
### Usual care (control arm)
The participants in the waiting-list (control) arm will receive usual care and will be offered PR after completing 12 weeks of follow-up. There are currently no guidelines for the clinical management of PTBLD both locally and internationally. Usual care will be optimised where possible and will include the following: frontal chest

radiograph, spirometry to screen for airway diseases, inhalational therapies for airway disease amenable to treatment (where appropriate), antibiotic and systemic glucocorticoid therapy for infective exacerbations (where appropriate), and verbal advice to quit smoking and reduce exposure to biomass smoke. According to local practice, all post-TB patients with significant post-bronchodilator response on Spirometry (at least 12% and 200mls increase in forced expiratory volume in 1s (FEV1)) are managed with a combination of inhaled corticosteroids and long-acting beta-agonists, while those with fixed airflow obstruction (postbronchodilator FEV1/forced vital capacity ratio of less than 0.70) are managed with long acting bronchodilators. PR will be offered as an adjunctive non-pharmacological treatment as recommended by international guidelines.[19]

## PR (trial intervention arm)

In addition to usual care described above, participants in the intervention arm will receive PR. PR will consist of a 6-week programme offered to a group of up to 12 participants, with sessions occurring twice weekly for at least 2 hours (approximately 1 hour for education and 1 hour for exercise).

## Warm-up and cool-down

Before starting exercises, participants will be taken through a group warm up session, followed by a cool down session at the end of exercises, each lasting 10–15 min. Warm up is aimed at readying the body for both the physical aspects of performance (increased blood flow and muscle temperature) and mental readiness for exercise while cool down session facilitates a smoother decline in temperature and blood flow[20] Both warm up and cool down will consist of stretching and flexibility exercises during which participants will perform both upper and lower body flexibility exercises, held for 10–15 s each (including stretching of major muscle groups such as the calves, hamstrings, quadriceps and biceps, as well as range of motion exercises for the neck, shoulders and trunk), 2 days/week.[13] The cool down session has the same activities of warm-up (online supplemental table 1) but performed at a slower pace.

## Endurance training

Each participant will go through two stations of endurance exercise; load-adjustable stationary cycling and ground-based walking stations. We shall employ an intensity of continuous exercise at each station for 10 min or until a Borg dyspnoea score of 4–6 (moderate to (very) severe) is attained.[21 22] Participants who may have difficulty in sustaining continuous high-intensity exercise will have interspersed periods of rest or lower intensity exercise to maximise the benefits of exercise training.[13] The walking exercise regime will be individually prescribed to participants based around their performance in the ISWT. Participants will be encouraged to walk at 85% of their maximal ISWT walking speed.[23]

---

**Box 1  Education content of the Global RECHARGE Pulmonary Rehabilitation programme**

1. Normal anatomy and physiology of the lungs.
2. Pathophysiology of chronic lung disease.
3. Tuberculosis and how it causes lung damage.
4. Coping with chronic lung disease and coping with stress.
5. Avoidance of risk factors for chronic lung disease.
6. Early recognition and treatment of exacerbations.
7. Strategies for managing breathlessness.
8. Energy conservation during activities of daily living.
9. Role and rationale for medications and devices.
10. Benefit of exercise and physical activities.
11. Healthy food intake.
12. Secretion clearance techniques.

---

## Strength training

Each participant will go through two stations for strengthening upper limb muscles (pull-ups and biceps curls) and two for strengthening lower limb muscles (sit-to-stand and step-up exercises). Each of the stations will include 3 sets of 8–12 repetitions. Participants will be asked to continue doing both endurance and resistance exercises at home, unsupervised.

## Education sessions

A dedicated education session will be conducted at the start of each class, before the exercise regimes (box 1; 12 sessions in total).

## Study outcomes
### Primary outcome

The primary outcome is change in walking distance measured by the ISWT from preintervention to postintervention. A group change of at least 35 m is considered clinically important.[18]

### Incremental Shuttle Walking Test

The ISWT is frequently used as an outcome measure for PR.[24] Improvement in walking distance of 35 m during the post-PR shuttle test, measured from baseline (pre-PR) using the ISWT is considered a clinically important difference.[18] The ISWT requires the patient to walk up and down a 10 m course, identified by two cones inset 0.5 m from either end to avoid the need for abrupt changes in direction. The speed at which the patient walks is dictated by an audio signal played on an audio device. Each participant will receive standardised instructions to: 'Walk at a steady pace, aiming to turn around when you hear the signal. You should continue to walk until you feel that you are unable to maintain the required speed without becoming unduly breathless'.[25] To ensure the learning effect is accounted for, a practice ISWT will be performed and the participant will receive encouragement from the physiotherapist throughout the test in an effort to increase the distance one can walk. The test is terminated when either (1) the patient indicates that they are unable to continue, (2) if the operator determines that the

patient is not fit to continue or (3) the operator assesses that the patient was unable to sustain the speed and cover the distance to the cone prior to the beep sounding.[25]

### Secondary outcomes

All secondary outcomes will be compared between the PR and usual care arms from baseline to 6-week and 12-week follow-ups.

Health questionnaires will be administered including COPD assessment test (CAT), Clinical COPD questionnaire (CCQ), Hospital Anxiety and Depression Scale (HADS), Patient Health Questionnaire (PHQ-9), Work Productivity and Activity Impairment (WPAI), and European Quality of Life 5-Dimensions (EQ-5D-5L). PR-specific measurements will include the ISWT, Endurance Shuttle Walking Test (ESWT), mid upper arm circumference and sit-to-stand test.

### Respiratory symptoms

The CCQ is a simple 10-time validated HRQoL questionnaire with good psychometric properties.[26] It consists of 10 items, each scored between 0 and 6, divided into three domains (symptoms, functional, mental), with higher scores representing worse HRQoL. The CCQ is responsive to PR with an estimated minimal important improvement of 0.4.[27]

The CAT is a validated, self-administered, short and simple questionnaire that measures HRQoL.[28] The CAT consists of eight items, each scored between 0 and 5 scored with a range of 0–40; scores of 0–10, 11–20, 21–30, 31–40 representing mild, moderate, severe or very severe negative impact on HRQoL, respectively. The CAT is responsive to the effects of PR with an estimated minimal clinically important difference (MCID) of 2 points.[29]

### Psychological well-being

The HADS questionnaire is a validated, easy to use screening tool for anxiety and depression symptoms in a hospital outpatient setting.[30] The self-report rating scale is composed of 14 items with two 7-item subscales (HADS-Anxiety and HADS-Depression), both ranging from 0 to 21 with higher scores indicating more severe distress. The HADS is responsive to PR with estimated MCID of 2 points on each subscale.[31 32]

The PHQ-9 is a nine item, validated, short, self-administered and positively worded questionnaire designed to measure the severity of depression over the last 2 weeks.[33] The total score ranges from 0 to 27, with high scores indicating high depression, specifically; no depression (0–4), mild (5–9), moderate (10–14), moderately severe (15–19) or severe depression (20–27).[20–27 33] The PHQ-9 has an estimated MCID of 5 points.[34]

### Work productivity and impairment

The WPAI questionnaire is a validated instrument to measure impairments in work and activities, both paid and unpaid. The WPAI self-administered questionnaire measures time missed from work, impairment of work and regular activities due to overall health and symptoms,

during the past 7 days.[35] We have added two follow-up supplementary questions, following the WPAI format, to measure productivity with respect to regular household duties in low-resource settings.

### Health-related quality of life

The EQ-5D-5L questionnaire is a standardised questionnaire, developed to measure of health outcomes and defines health in terms of five dimensions: mobility, self-care, usual activities, pain or discomfort and anxiety or depression.[36] The questionnaire also contains a visual analogue scale. The EQ-5D-5L will be used to calculate patient costs per quality-adjusted life-year. EQ-5D-5L is responsive to change following PR, with a MCID of 0.05 (utility index) and 7.0 (Visual Analogue Scale).[37]

### Exercise capacity/ physical function

The five-repetition sit-to-stand test (FTSTS) is a commonly used functional performance measure of lower-limb strength.[38] The FTSTS measures the time taken to stand five times from a sitting position as rapidly as possible. The FTSTS is reliable, valid and responsive to PR with an estimated MCID of 1.7 s.[39]

The MRC dyspnoea scale is a 5-point self-administered questionnaire based on the sensation of breathing difficulty experienced by the patient during daily life activities. The questionnaire is short, easy to use and has grades ranging from 1 (none) to 5 (almost compete incapacity), with high grades indicating high perceived respiratory disability.[40] The MRC dyspnoea scale is responsive to PR with estimated MCID of 1 points.[41 42]

The ESWT is a constant-load exercise test which measures the ability of the participant to sustain a given submaximal exercise capacity; the participant aims to walk at 85% of their maximal ISWT walking speed.[23] The ESWT is frequently used as an exercise tolerance outcome measure for PR. The endpoint of the test is the time the participant walks at the constant endurance speed. The test consists of prerecorded audio signals at different frequencies giving a total of 16 walking speeds. The ESWT is responsive to PR with MCID following a 6 week PR programme between 174 and 279 s.[43]

### Physical activity

Participants will be asked to wear an ActiGraph wGT3X-BT activity monitor (ActiGraph, Pensacola, Florida, USA), able to detect a range of PA intensities.[44] Participants will be instructed to wear the PA monitor on the right anterior hip during waking hours for 1 week prior to attending PR (preintervention) and for 1 week prior to their postintervention assessment (online supplemental table 2). Written instructions to follow will be provided to the participants prior to wearing and using the PA monitors.

### Cost/benefit analysis

The cost of starting and running a PR programme will include single and recurrent costs (table 2). Single payments will include the necessary costs needed to set

**Table 2** Table showing the variables used to calculate fixed and recurrent costs (not an exhaustive list)

| Fixed costs | Recurrent costs |
|---|---|
| ► Electrical equipment (laptop, printer, projector)<br>► Equipment for PR (weights, treadmill, cycle ergometer, country-specific equipment, step-up box, chairs)<br>► Equipment for shuttle walking tests (cones, licences, stop watches, tape measure, electrical equipment to play audio)<br>► Equipment for PR assessment (height stadiometer, weight scales, sphygmomanometer, pulse oximeter, spirometer, calibration syringe, country-specific equipment)<br>► Additional safety equipment (blood glucose monitor, oxygen cylinder holder)<br>► Miscellaneous (filing cabinets, storage units, questionnaire translations, questionnaire licences, staff uniform)<br>► Staff time (creating core PR content including educational material, exercise diaries and other necessary paperwork) | ► Staff time to conduct PR (assessment at baseline and discharge, conduct PR classes, telephone calls and data entry)<br>► Disposable equipment (for blood glucose monitor, spirometer mouthpieces, nose-clips)<br>► Servicing costs (spirometer, PR equipment, specifically cycle ergometers)<br>► Miscellaneous (oxygen cylinders, questionnaire licences, stationery (paper)<br>► Patient costs (transport and meals) |

PR, pulmonary rehabilitation.

up and run PR. Recurrent costs refer to any item with a life expectancy of ≤1 year (eg, disposable materials).[45] The fixed costs will be captured prior to enrolling the first participant into the PR programme and the recurrent costs will be collected at the mid-stage of recruitment. The average fixed and recurrent costs will be calculated separately.

### Patient and public involvement
Adults with CRDs tell us how they are greatly troubled by breathlessness and express interest in attending a programme that can help better manage their condition. They express interest in attending a hospital-based programme that allows them to interact with fellow patients. They additionally tell us how the PR programme should be delivered. We have also set up a patient and public involvement group at MLI that will meet regularly, and assist with disseminating results following the study.

### Data analysis
All randomised participants will be included in the intention to treat analysis population. The primary efficacy analysis will be based on both per protocol and modified intention-to-treat populations. For the primary analysis, the differences in the primary outcome (walking distance on the ISWT) with the corresponding two-sided 95% CI and p value will be estimated using a stratified analysis; a $p < 0.05$ will be the measure for statistical significance. Predictive analytics software (SPSS[16]) will be used to analyse the data. Continuous data will be presented as mean and SD or median and IQRs, while categorical data will be presented as frequencies and percentages. All data will be assessed for normality and appropriate parametric and non-parametric tests will be used. Categorical variables between the two treatment groups will be compared using $\chi^2$ and Fisher's exact test as appropriate. Continuous variables will be compared using t-test for normally distributed data and Mann-Whitney-U test for non-normally distributed data. Any baseline differences will be adjusted for. Both intention-to-treat and

per-protocol analyses will be conducted after imputing any missing data. There will be no formal interim analysis of data. The final analysis will be performed when all the 114 participants have completed the last study related visit or previously withdrawn from the trial. We will fit linear mixed models for both per-protocol and intention-to-treat analyses.

### Data management
An independent data monitoring committee will be established at the University of Leicester, UK to review high level safety data (serious adverse events and adverse events) at least quarterly, and as needed on an ad hoc basis to ensure the continuing safety of the participants enrolled in this study.

All data collected during the trial will be entered into the Research Electronic Data Capture[46 47] with access via a secure password protected web interface hosted by the University of Leicester, UK. Study participants will be assigned a study-specific identification code.

## ETHICS AND DISSEMINATION
The study received ethical approvals from the University of Leicester research ethics committee (UK) (Ref No. 22349) and locally from the Mulago Hospital Research and Ethics Committee (MHREC1478), Kampala, Uganda as well as the Uganda National Council for Science and Technology (SS5105).

### Confidentiality
The confidentiality of all participants will be protected to the fullest extent possible. All patient information will be kept secure and will be available only to the treatment staff and representatives of the sponsors, regulators and ethics committees.

All participants will be provided with a unique identification number which will be recorded in the participant enrolment log and stored in a secure place. Study participants will not be identified by name on any case report

form, email or on any other documentation sent to the central database and will not be reported by name in any report, presentation or publication resulting from data collected in this study. Participants' data/specimens will be identified by study number or hospital number only.

## Dissemination

Results of the study will be published in peer-reviewed journals and findings disseminated at appropriate local, regional and international scientific meetings and conferences. Social media will be used to disseminate information and summaries of results to a wider public domain. Furthermore, a participant dissemination meeting will be held following this trial, in which study participants will receive a summary of the findings.

## COVID-19 provisions

Modifications will be made to the delivery of the PR programme due to the COVID-19 pandemic. The PR room will be reorganised to allow for social distancing (minimum 2 m) for both study staff and study participants. The maximum number of participants participating in the PR session will be reduced from 12 to 8 to ensure social distancing between participants. Before accessing the PR room, all participants and staff will be required to undergo temperature measurement using a hand-held non-contact thermometer, wash hands with soap or alcohol-based hand sanitiser. All participants will be provided with face masks during PR sessions. All surfaces inside the PR room will be disinfected before and after every PR session. PR sessions will be conducted in the morning hours to allow participants travel back home in time before the evening rush hour and the standard operating procedure for data collection will be modified ensure 2-metre distancing between the study staff and study participant. Study participants will undergo COVID-19 testing before starting PR and as needed during the hospital based sessions. All study staff will be required to wear N95 masks at all times and will undergo COVID-19 training with emphasis on infection prevention and control, and screening study participants for signs and symptoms of the disease.

**Author affiliations**
[1]Makerere University Lung Institute, Kampala, Uganda
[2]Department of Respiratory Sciences, University of Leicester, Leicester, UK
[3]Centre for Exercise and Rehabilitation Science (CERS), NIHR Leicester Biomedical Research Centre – Respiratory, University Hospitals of Leicester NHS Trust, Leicester, UK
[4]Department of Physiotherapy, Mulago National Referral Hospital, Kampala, Uganda
[5]Faculty of Health, University of Plymouth, Plymouth, UK
[6]NIHR Leicester Biomedical Research Centre, University Hospitals of Leicester NHS Trust, Leicester, UK
[7]Department of Economics, University of Sheffield, Sheffield, UK

**Contributors** SJS is the principal investigator of the Global RECHARGE project while BK is the in-country principal investigator. WK, MO, AVJ, RK, RBM, AM and RCF have been involved in drafting the work and revising it critically for important intellectual content. AB, JR, MS and JAM have substantially contributed to the development of the intervention and the design of the trial. All authors have revised the content and approved the final version to be published.

**Funding** This research was funded by the National Institute for Health Research (NIHR) (17/63/20) using UK aid from the UK Government to support global health research.

**Disclaimer** The views expressed in this publication are those of the author(s) and not necessarily those of the NIHR or the UK Department of Health and Social Care.

**Competing interests** None declared.

**Patient consent for publication** Not required.

**Provenance and peer review** Not commissioned; externally peer reviewed.

**ORCID iDs**
Winceslaus Katagira http://orcid.org/0000-0003-4622-191X
Mark W. Orme http://orcid.org/0000-0003-4678-6574
Amy V. Jones http://orcid.org/0000-0001-6565-8645

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
