## [Reviewer comments · BMJ Open]

ARTICLE DETAILS

TITLE (PROVISIONAL)	A study protocol for a randomised controlled trial assessing the impact of Pulmonary Rehabilitation on maximal exercise capacity for adults living with Post-TB lung disease: Global RECHARGE Uganda
AUTHORS	Katagira, Wincelous; Orme, Mark; Jones, Amy; Kasiita, Richard; Rupert, Jones; Barton, Andy; Miah, Ruhme; Manise, Adrian; Matheson, Jesse; Free, Robert; Steiner, michael; Kirenga, Bruce; Singh, Sally

VERSION 1 – REVIEW

REVIEWER	Chen, Chin-Ming Chi Mei Medical Center, Intensive Care Medicine
REVIEW RETURNED	25-Jan-2021

GENERAL COMMENTS	The protocol entitled "A study protocol for a randomised controlled trial assessing the impact of Pulmonary Rehabilitation on maximal exercise capacity for adults living with Post-TB lung disease: Global RECHARGE Uganda" was reviewed. This research was funded by NIHR using UK aid from the UK Government to support global health research. And this randomized waiting-list controlled trial with blinded outcome measures, is conducted to compare 6-week pulmonary rehabilitation (PR) versus usual care for TB survivors in a teaching and clinical research hospital in Uganda, the sub-Saharan Africa. A total of 114 participants will be randomized (1:1) to receive either usual care (waiting-list) or PR. The primary objective of this trial is to assess the impact of PR on the maximal exercise capacity using the incremental shuttle walking test (ISWT), and the secondary objectives include assessing the quality of life and other outcomes, and to conduct a cost-benefit analysis of PR. The authors also do some modifications of delivery of PR program due to COVID-19 pandemic. The protocol was well written, with the possibility of dissemination of PR program on TB survivors in low-and middle income countries (LMIC) as in Africa. I had some questions about the protocol: 1. About the recruitment of smear positive TB, how to define the true TB infection rather than non-tuberculosis mycobacterium (NTM)? The positive TB culture or a positive Xpert MTB/RIF from broth culture? It should be defined clearly.2. About those patients included in the protocol, if they are unable to do PR program, such as admission to hospital, recurrent TB infection, or other complication post TB infection, what will they do? Withdrawal from the study? Or wait until their recovery and resume soon?
---

REVIEWER	Byrne, Anthony University of New South Wales, Medicine
REVIEW RETURNED	04-Feb-2021

GENERAL COMMENTS	The authors are to be congratulated on this protocol to address an important topic relevant to many people globally. I have some suggestions and questions.  1. The primary outcome is a change in walk distance. Is this outcome known to correlate with clinically significant endpoints? 2. Sample size. Is the stated difference in primary outcome clinically significant? Why was only 80% power (not 90%) used? The sample size of 57 per group (114 total) and drop out rate of 30% doesn't appear consistent with the stated total numbers of participants. Please clarify. 3. The secondary outcomes are all important and clearly defined. Is there any estimation of the ability of the planned sample size to detect differences in these outcomes? The mMRC is stated in the methods but not specifically in the secondary outcomes. Given the entry criteria for inclusion is > or =2 this would be important to specifically include in the secondary outcomes 4. Inclusion. How are PTBLD patients defined? It appears that this cohort is well defined and known to the authors but I am unclear from the protocol who is included (other than treated TB with no active disease and with current breathlessness). 5. Time since TB should be included as a defined variable. The response to PR may be different for those recently completed TB treatment to those years after TB treatment completion. 6. Pharmacotherapy with bronchodilators (if abnormal spirometry) and antibiotics (if exacerbation) are mentioned briefly. I would think that a current exacerbation requiring antibiotics should preclude/delay enrollment ? As the authors state, there are no defined guidelines for prescription of bronchodilators in this cohort outside of the GOLD guidelines for COPD. How will inhalers be prescribed? This is a potential confounder to the intervention so requires careful thought. 7. Smoking. Are specific (cigarette) smoking interventions planned for both groups? Is smoking status recorded (I may have missed it). 8. If there is a positive effect of PR, will it be offered to participants in the control arm after the study.
---

VERSION 1 – AUTHOR RESPONSE

Reviewer: 1
Dr. Chin-Ming Chen, Chi Mei Medical Center

The protocol entitled "A study protocol for a randomised controlled trial assessing the impact of

Pulmonary Rehabilitation on maximal exercise capacity for adults living with Post-TB lung disease: Global RECHARGE Uganda” was reviewed. This research was funded by NIHR using UK aid from the UK Government to support global health research. And this randomized waiting-list controlled trial with blinded outcome measures, is conducted to compare 6-week pulmonary rehabilitation (PR) versus usual care for TB survivors in a teaching and clinical research hospital in Uganda, the sub-Saharan Africa. A total of 114 participants will be randomized (1:1) to receive either usual care (waiting-list) or PR. The primary objective of this trial is to assess the impact of PR on the maximal exercise capacity using the incremental shuttle walking test (ISWT), and the secondary objectives include assessing the quality of life and other outcomes, and to conduct a cost-benefit analysis of PR. The authors also do some modifications of delivery of PR program due to COVID-19 pandemic. The protocol was well written, with the possibility of dissemination of PR program on TB survivors in low-and middle income countries (LMIC) as in Africa. I had some questions about the protocol:

1. About the recruitment of smear positive TB, how to define the true TB infection rather than non-tuberculosis mycobacterium (NTM)? The positive TB culture or a positive Xpert MTB/RIF from broth culture? It should be defined clearly.

We wish to clarify that the trial will only recruit post-TB lung disease (PTBLD) patients (i.e, participants who recovered from smear-positive TB). The study will exclude patients with a history of concurrent (positive Xpert MTB/RIF test or those on treatment for smear positive TB) or prior TB treatment within the previous six months.

We underline this previously included information, which is found in the inclusion criteria, on pages 5 and 6, lines 158-163.

A patient with PTBLD is eligible for the trial if they meet all of the following criteria: aged ≥ 18 years, willing and able to provide written informed consent (signed or witnessed consent if the patient is illiterate), a documented past history of smear positive pulmonary TB with treatment completed ≥ 6 months prior to study enrolment, a negative Xpert MTB/RIF assay for Mycobacterium tuberculosis at the time of study enrolment, and report a Medical Research Council (MRC) dyspnoea grade ≥ 2 .

2. About those patients included in the protocol, if they are unable to do PR program, such as admission to hospital, recurrent TB infection, or other complication post TB infection, what will they do? Withdrawal from the study? Or wait until their recovery and resume soon?

Participants randomized to the Pulmonary Rehabilitation arm but fail to complete the trial intervention for any reason will still be included in the Intention to treat analysis. The study is powered to allow up to 30% of drop-outs. However, if the hospitalization is for a short time, participants receiving the intervention (PR) will resume the trial.

Reviewer 2:

Dr. Anthony Byrne, University of New South Wales

The authors are to be congratulated on this protocol to address an important topic relevant to many people globally. I have some suggestions and questions.

1. The primary outcome is a change in walk distance. Is this outcome known to correlate with clinically significant endpoints?

Yes, but we note that this is not explicitly stated. We have added the sentence on page 10, lines 279-281 to address this omission:

“The ISWT is frequently used as an outcome measure for PR. Improvement in walking distance of 35m during the post-PR shuttle test, measured from baseline (pre-PR) using the ISWT is considered a clinically important difference.”

2. Sample size. Is the stated difference in primary outcome clinically significant? Why was only 80%

power (not 90%) used?

We appreciate this question and respond as follows; 80% power is appropriate for a study of this nature and offers clinical value whilst remaining feasible for the resources available to us. The sample size, power, and all the rest are determined by what can be realistically and practically obtained in accordance with the MCID for the primary outcome.

The sample size of 57 per group (114 total) and dropout rate of 30% doesn't appear consistent with the stated total numbers of participants. Please clarify.

We acknowledge this typing error. The 114 includes a 30% dropout rate. We have modified the sentence on page 7, lines 209-215 to read as follows;

“Based on a trial sample size of 40 participants in each of the treatment and control groups, a 2-sided 5% significance level and a statistical power of 80%, the clinically important change in ISWT of 35m will also be statistically significant. Our recent feasibility study [15] was used to obtain an estimate of the pooled standard deviation for the power calculation. Conservatively assuming up to 30% loss to follow-up at 6-weeks, a total of 114 participants are required to be recruited and randomised (1:1) to each arm (PR: 57 participants or waiting list: 57 participants).”

3. The secondary outcomes are all important and clearly defined. Is there any estimation of the ability of the planned sample size to detect differences in these outcomes?

The study is powered to detect clinically important differences in the ISWT but it is also sufficient for secondary outcomes, including MCIDs for the CCQ and CAT.

The mMRC is stated in the methods but not specifically in the secondary outcomes. Given the entry criteria for inclusion is $>$ or $=2$ this would be important to specifically include in the secondary outcomes.

We agree with this comment and confirm the MRC dyspnoea scale as a secondary outcome, as highlighted on page 12 lines 344-348 under “exercise capacity/physical function”;

“The MRC dyspnoea scale is a 5-point self-administered questionnaire based on the sensation of breathing difficulty experienced by the patient during daily life activities. The questionnaire is short, easy to use and has grades ranging from 1 (none) to 5 (almost complete incapacity), with high grades indicating high perceived respiratory disability. The MRC dyspnoea scale is responsive to PR with estimated MCID of 1 points.”

4. Inclusion. How are PTBLD patients defined? It appears that this cohort is well defined and known to the authors but I am unclear from the protocol who is included (other than treated TB with no active disease and with current breathlessness).

We acknowledge that PTBLD is not explicitly defined and that this would be helpful to readers. We have added the study definition of PTBLD on page 5, lines 140-143;

“In this study, a patient is considered to have post-TB lung disease (PTBLD) if they successfully completed treatment for microbiologically confirmed Pulmonary TB but continue to experience chronic respiratory symptoms with radiological evidence of lung parenchymal damage.”

5. Time since TB should be included as a defined variable. The response to PR may be different for those recently completed TB treatment to those years after TB treatment completion.

We agree with this suggestion and include this as part of our baseline data collection, and we will report this data.

6. Pharmacotherapy with bronchodilators (if abnormal spirometry) and antibiotics (if exacerbation) are mentioned briefly. I would think that a current exacerbation requiring antibiotics should preclude/delay enrollment? As the authors state, there are no defined guidelines for prescription of bronchodilators in this cohort outside of the GOLD guidelines for COPD. How will inhalers be prescribed? This is a potential confounder to the intervention so requires careful thought.

We appreciate this question and respond as follows on page 8, lines 134-140;

“According to local practice, all post-TB patients with significant post-bronchodilator response on Spirometry (at least 12% and 200mls increase in forced expiratory volume in 1 second (FEV1)) are managed with a combination of inhaled corticosteroids and long-acting beta-agonists, while those with fixed airflow obstruction (post-bronchodilator FEV1/forced vital capacity (FVC) ratio of less than 0.70) are managed with long acting bronchodilators. PR will be offered as an adjunctive non-pharmacological treatment as recommended by international guidelines.”

We agree that post-discharge patients previously admitted with an infective exacerbation will be enrolled 6 weeks after hospital discharge.

7. Smoking. Are specific (cigarette) smoking interventions planned for both groups? Is smoking status recorded (I may have missed it).

Yes. Smoking status will be collected for all participants during baseline assessment and all current smokers will be advised to quit smoking as per local practice.

We have modified the sentence on page 8 line 233 to make this clearer.

We'd also like to add that the prevalence of tobacco smoking in Uganda is low with an estimated the prevalence of smoking in adults to be 12.9% in men and 0.6% in women according to the 2018 World Health Organization (WHO) Global Report on Trends in Prevalence of Tobacco Smoking 2000-2025.

8. If there is a positive effect of PR, will it be offered to participants in the control arm after the study.

Yes, we agree this is very important and we highlight this previously included information, which is found in the treatment arms, usual care (control arm) section, page 8, lines 227-228.

“The participants in the waiting-list (control) arm will receive usual care and will be offered PR after completing 12-weeks of follow-up.”

Figure 1: Study flow diagram also elaborates on this.

VERSION 2 – REVIEW

REVIEWER	Byrne, Anthony University of New South Wales, Medicine
REVIEW RETURNED	07-May-2021
GENERAL COMMENTS	The authors have adequately addressed my questions and concerns.